



# The SeaSWIR dataset

Knaeps Els[1], David Doxaran[2], Ana Dogliotti [3], Bouchra Nechad[4], Kevin Ruddick[4], Dries Raymaekers[1], Sindy Sterckx[1].

[1]Flemish Institute for Technological Research (VITO), Mol, 2400, Belgium

[2]Laboratoire d'Océanographie de Villefranche (LOV), Villefranche-sur-Mer, 06230, UMR 7093, CNRS/UPMC, France

[3]Instituto de Astronomía y Física del Espacio (IAFE), C1428ZAA,CONICET/UBA, Argentina

[4]Royal Belgian Institute for Natural Sciences (RBINS), Operational Directorate Natural Environments, Brussels, 1200, Belgium

*Correspondence to*: Els Knaeps (els.knaeps@vito.be)

**Abstract**

The SeaSWIR dataset consists of 137 ASD (Analytical Spectral Devices, Inc.) marine reflectances, 137 Total Suspended Matter (TSM) measurements and 97 turbidity measurements gathered at three turbid estuarine sites (Gironde, La Plata, Scheldt). The dataset is valuable because of the high quality measurements of the marine reflectance in the Short Wave InfraRed I region (SWIR-I: 1000-1200nm) and SWIR-II (1200-1300nm) and because of the wide range of TSM concentrations from 48 mg L-1 up to 1400 mg L-1 . The ASD measurements were gathered using a detailed measurement protocol and were subjected to a strict quality control. The SeaSWIR marine reflectance is characterized by low reflectance at short wavelengths (< 450 nm), peak reflectance values between 600 and 720 nm and significant contributions in the Near InfraRed (NIR) and SWIR-I parts of the spectrum.  Comparison of the ASD water reflectance with simultaneously acquired TRIOS reflectance revealed a correlation of 0.98 for short wavelengths (412, 490 and 555 nm) and 0.93 for long wavelengths (686, 780 and 865 nm).  The relationship between TSM and turbidity (for all sites) is linear with a correlation coefficient of 0.96.

The SeaSWIR dataset has been made publicly available (https://doi.org/10.1594/PANGAEA.886287).

**1.Introduction**

The SeaSWIR dataset (https://doi.org/10.1594/PANGAEA.886287) has been collected in the framework of the SeaSWIR project (Remote sensing of turbid waters in the Short Wave Infrared) to determine the variability of the marine reflectance in the Short Wave InfraRed (SWIR, 1000-2500 nm) as function of Total Suspended Matter (TSM) concentration and Turbidity (T). The marine reflectance in the SWIR has not been reported before because of specialised optical instrumentation is required to measure in the SWIR. More information on the SWIR marine reflectance is essential because several satellites have sensors with spectral bands in this range e.g. Sentinel-3 OLCI with a spectral band at 1020 nm, MODIS AQUA and TERRA with spectral bands at 1240, 1640 and 2130 nm, Visible Infrared Imaging Radiometer Suite (VIIRS) with spectral bands at 1240, 1378, 1610 and 2250 nm. Knowledge of the marine reflectance in these spectral bands will be used first of all to underpin or refute the assumptions of a black pixel (often assumed in atmospheric correction over  turbid waters). However, earlier publications using this dataset (Knaeps et al., 2012; Knaeps et al., 2015) have also shown the existence of a relationship between the SWIR marine reflectance at 1020 and 1070 and the TSM concentrations and have proposed a TSM algorithm for the SWIR bands. Such a SWIR TSM algorithm is particularly useful in coastal waters with extremely high turbidity where shorter bands in the NIR saturate.



In Knaeps et al. (2015) the SWIR region between 1000 and 1300 nm was further divided into two sub-regions based on the shape and magnitude of the pure water absorption. The SWIR–I region ranges from 1000 nm to 1200 nm, close to a local maximum in the pure water absorption. The SWIR-II region ranges from 1200 nm to 1300 nm where the pure water absorption reaches a value of 132.2 m$^{-1}$.

To measure marine reflectance in the SWIR-I and SWIR-II, a typical 'land' ASD (Analytical Spectral Devices, Inc.) spectroradiometer (FieldSpec) was used, recording data from 350 to 2500 nm. A measurement methodology was developed for the ASD to derive the marine reflectance from this single radiometer system and results were compared with marine reflectance measured using a three radiometer system (TRIOS). As the pure water absorption is high in the SWIR, only water with a high concentration of suspended particles will have a measurable reflectance. Therefore, test sites were chosen

with high to extremely high turbidity: the Scheldt river in Belgium, the Gironde estuary in France and Río de la Plata estuary in Argentina (Figure 1). Measurement campaigns were organized at all three sites, mainly on fixed pontoons along the rivers/estuaries.

The measurement campaigns resulted in a large dataset of 137 ASD reflectance spectra in the Visible/Near InfraRed (VNIR) and SWIR with corresponding TSM concentrations and turbidity. This paper describes the methodology for data acquisition

and processing and analyses the final reflectance, TSM concentrations and turbidity values. Additional quality control of the ASD reflectance is performed with simultaneously acquired TRIOS reflectance data.

**2.Sites**

Field data has been collected in three different estuaries with high to extremely high TSM concentrations: the Scheldt in Belgium, the Gironde in France and the Río de la Plata in Argentina (Figure 1). . These three estuaries were chosen because

of their high concentration range of suspended particles and diversity of particle composition/type. It was expected that the high concentration of TSM will result in a measurable water reflectance signal in the SWIR.:.

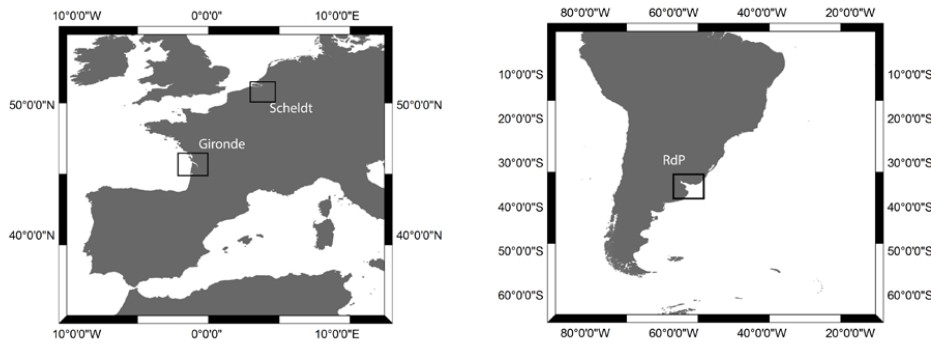

Figure 1: Location of the study sites: Scheldt (Belgium), Gironde (France), and Río de La Plata (Argentina).


The Scheldt River has its origin in France, flows through Belgium and reaches the North Sea in the Netherlands. It forms a macrotidal estuary leading to TSM concentrations within surface waters varying from a few to several hundreds mg L$^{-1}$ during one tidal cycle. The study area is located near the city and the harbor of Antwerp where there is a tide-dominated Maximum Turbidity Zone (MTZ). The tidal current induces re-suspension of local bottom sediments leading to high





suspended matter concentration (Chen *et al.*, 2005). The field sampling was performed on the pontoon Sint Anna near the city of Antwerp in 2010 and 2012 (Table 1).

The Gironde is a macrotidal estuary where the water optical properties are dominated by non-algal particles, *i.e.* by suspended sediments (silts and clays) delivered by the Garonne and Dordogne Rivers and trapped within the maximum

turbidity zone of the estuary. TSM concentrations typically vary from ten to four thousands mg $L^{-1}$ within surface waters, mainly depending on fortnightly tidal cycles (Doxaran *et al.*, 2009). Field sampling was performed on a vessel and on the pontoons of Blaye and Pauillac in 2012 and 2013 (Table 1).

The Río de la Plata is a shallow (< 20 m) and large-scale estuary which drains the second largest basin in South America. High values of TSM have been reported in this region, with mean values ranging from 100 to 300 mg $L^{-1}$ and extreme

concentrations around 1000 mg $L^{-1}$ in the Maximum Turbidity Zone (Dogliotti et al. 2014). This estuary has been identified by a number of investigators (Shi and Wang, 2009; Doerffer, 2006; Moore *et al.*, 2010; Doron *et al.*, 2011) as a very bright target which can be used for evaluating the performance of atmospheric correction algorithms, so far tested without any *in situ* data (Dogliotti et al. 2011). Field sampling was performed on the Club de Pescadores Pier in Buenos Aires in 2012 (Table 1).

Most of the measurements from the SeaSWIR dataset were collected from fixed pontoons along the side of the rivers/estuaries. These pontoons allow to capture a large variation of TSM concentration and T during the daily tidal cycle and allow to make high quality reflectance measurements. In particular, the verticality of the irradiance sensor or horizontality of the ASD reflectance plaque can be much better controlled than on a ship. The range of azimuth viewing

angles on such pontoons can be limited because of shadow/reflection by the structure itself.

### 3. Data campaigns

Five sampling campaigns and in total 23 measurement days were organized between 2010 and 2013 on all test sites (Table 1).


In 2010, the first field campaigns were organized at the Sint Anna pontoon on the Scheldt River (Antwerpen, Belgium). The campaign included two days of measurement, one day in July and one day in October. The campaign was repeated in the same pontoon in June 2012 with a 2-day campaign.

Later in the same month, an extensive field campaign was organized in the Gironde estuary (France) with 3 days of measurements from a vessel and 3 days of measurements from fixed pontoons:

- 1st day (11/06/2012) onboard the research vessel: fixed location close to the river mouth, 85 km downstream Bordeaux
- 2nd day (12/06/2012) onboard the research vessel: fixed location 67 km downstream Bordeaux
- 3rd day (13/06/2012) onboard the research vessel: fixed location 30 km downstream Bordeaux
- 4th day (14/06/2012) from a pontoon located in Blaye, right shore of the estuary
- 5th day (15/06/2012) from the harbour wall located in Pauillac, left shore of the estuary

During two weeks (12-23 November 2012) a 9-day campaign was organized at the Club de Pescadores Pier at the Río de la Plata in Buenos Aires (Argentina).




Finally a last campaign was organized in the Gironde in August 2013. A 5-day sampling campaign was organized at the pontoons of Pauillac and Blaye.

All measurements are listed in the tables below.

**Table 1: Date and location of *in situ* campaigns**

| Campaign | Date | Station | Latitude (deg) | Longitude (deg) |
|---|---|---|---|---|
| Scheldt 2010 | 15/07/2010 26/10/2010 | Sint Anna pontoon | 51,234 | 4,397 |
| Scheldt 2012 | 2/06/2012 5/06/2012 | Sint Anna pontoon | 51,234 | 4,397 |
| Gironde 2012 | 11-13/06/2012 | onboard the research vessel | 45,517 | -1,950 |
| Gironde 2012 | 14/06/2012 | Blaye pontoon | 45,125 | -0,667 |
| Gironde 2012 | 15-16/06/2012 | Paulliac harbour wall | 45,198 | -0,742 |
| Gironde 2013 | 16/08/2013 | Blaye pontoon | 45,125 | -0,667 |
| Gironde 2013 | 12-15/08/2013 | Paulliac harbour wall | 45,198 | -0,742 |
| La Plata 2012 | 14-16/11/2012 19/11/2012 21-23/11/2012 | Club de Pescadores Pier | -34,561 | -58,399 |

## 4. Data collection

At all test sites concomitant water reflectance, turbidity and TSM measurements were made
(https://doi.org/10.1594/PANGAEA.886287). Table 2 provides an overview of the measurements performed at each campaign and station. The reported number of measurements refers to the remaining number of measurements after quality control. The measurement methodology is described in detail below.

**Table 2: Data collection. Campaign site location and year, type of platform used, number of radiometry (using ASD and Trios**
**radiometers), Total suspended Matter, and Turbidity measurements collected.**

| Campaign | station | ASD | TRIOS | TSM | T |
|---|---|---|---|---|---|
| Scheldt2010 | Sint Anna pontoon | 15 | 10 | 15 | 15 |
| Scheldt2012 | Sint Anna pontoon | 17 | 38 | 17 | 17 |
| Gironde2012 | vessel | 15 | - | 15 | 0 |
| Gironde2012 | Blaye pontoon | 15 | - | 15 | 0 |
| Gironde2012 | Paulliac pontoon | 15 | 16 | 15 | 5 |
| La Plata 2012 | Club de Pescadores Pier | 33 | 74 | 33 | 33 |
| Gironde2013 | Blaye pontoon | 4 | 15 | 4 | 4 |
| Gironde2013 | Paulliac pontoon | 23 | 47 | 23 | 23 |

### 4.1 ASD water reflectance

Water reflectance ($R_w$) between 350-2500 nm was measured with an ASD Fieldspec FR spectrometer. The ASD
spectrometer measures above the water light that is backscattered from the water and the light reflected at the air-water interface in the Visible/Near Infrared (VNIR, 350- 1050 nm) and the Short-Wave Infrared (SWIR, 900 – 2500 nm) parts of





the spectrum. The VNIR spectrometer has a spectral resolution of approximately 3 nm at around 700 nm. The spectral resolution in the SWIR varies between 10 nm and 12 nm. All measurements were performed using a   fore optics lens limiting the field-of-view to 8 degrees. Each individual measurement that the ASD makes is an average of 40 scans.

As the ASD has only one radiometer, three consecutive measurements have to be made to obtain the downwelling irradiance,
the total upwelling  radiance and  downwelling sky radiance.  To do so, the instrument was mounted on a steel frame which can be rotated easily keeping a 40° measurement angle from the nadir.

Measurements were performed under stable cloudy and sunny skies. Patchy clouds and highly variable light conditions were avoided. The three individual measurements are performed as follows:

The downwelling irradiance above the surface ($E_d(0+)$) was determined using an almost 100% reflecting Spectralon® reference panel (Analytical Spectral Devices, Inc.). The Spectralon panel was placed perfectly horizontal and objects were avoided within the hemisphere to prevent shadows. Figure 2 shows the instrument set-up on the pontoon with the spectralon panel in light blue mounted on a tripod and the ASD in black pointing towards the spectralon panel. The measured signal is
the downwelling radiance reflected by the plaque ($L_{dspec}$) which can be used to determine $E_d(0+)$ (Doxaran *et al*., 2004).

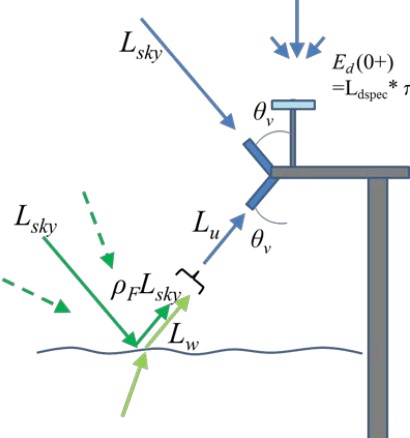

**Figure 2: Instrument set-up for $E_d(0+)$ measurement**

The total upwelling radiance from the water ($L_u(a)$) (i.e. from the water and from the air-sea interface) was measured by pointing the sensor at the water surface at 40° from nadir ($\theta_v$ in Figure 3), maintaining an azimuth of 90° or 135° from the solar plane, depending on the pontoon orientation with respect to the sun (Figure 3).

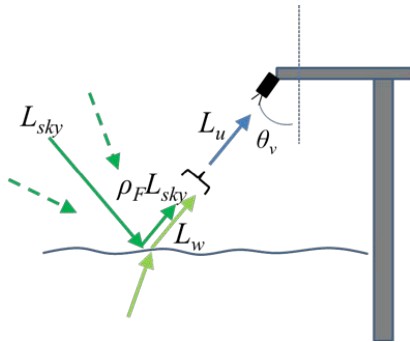




**Figure 3: Instrument set-up for $L_u(a)$ measurement**

The downwelling sky radiance ($L_{sky}(a)$) was measured by pointing the sensor towards the sky at a zenith angle of 40° ($\theta_v$ in Figure 4), to account for the skylight reflection, maintaining the exact same azimuth angle from the solar plane.

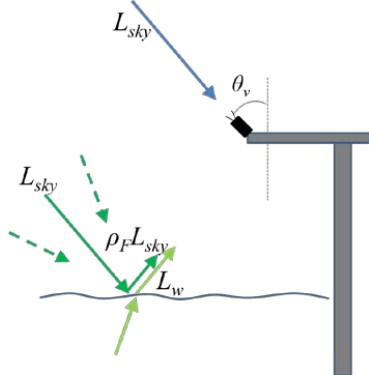

**Figure 4: Instrument set-up for $L_{sky}$ measurement**

A measurement procedure was defined of 21 consecutive measurements which is preceded by the optimization of the spectrometer integration time. Optimization is performed by pointing the optic head of the spectrometer towards the center of the Spectralon panel. Optimization results in automatic setting of Gains and Offsets for the SWIR detector, automatic setting of the Integration Time value for VNIR detector and automatic Dark Current collection. The instrument was re-optimized after change in lighting conditions . Then the  measurement sequence is as follows:

$E_{d1}(0+)$, $L_{u1}(a)$, $L_{sky1}(a)$, $L_{u2}(a)$, $L_{sky2}(a)$, $L_{u3}(a)$, $L_{sky3}(a)$

$E_{d2}(0+)$, $L_{u4}(a)$, $L_{sky4}(a)$, $L_{u5}(a)$, $L_{sky5}(a)$, $L_{u6}(a)$, $L_{sky6}(a)$

$E_{d3}(0+)$, $L_{u7}(a)$, $L_{sky7}(a)$, $L_{u8}(a)$, $L_{sky8}(a)$, $L_{u9}(a)$, $L_{sky9}(a)$

Each spectrum is saved in digital numbers which are 16-bit numbers between 0 and 65,535. While performing the measurements, start and end time of a measurement sequence was written down, as well as information on the atmospheric conditions and state of the sea surface. The latter were used in the quality control of the data.

The water reflectance ($R_w$) is calculated using the following equation (Mobley, 1999):

$L_{u\_avg}(\lambda)= (L_{u1}(\lambda)+ L_{u2}(\lambda)+ L_{u3}(\lambda))/3$

$L_{sky\_avg}(\lambda) = (L_{sky1}(\lambda)+ L_{sky2}(\lambda)+ L_{sky3}(\lambda))/3$

$R_{w1}(\lambda) = \pi(L_{u\_avg}(\lambda)-\rho_{as}L_{sky\_avg}(\lambda))/E_{d1}(\lambda)$

$\rho_{as}$ is the air-sea interface reflection coefficient. For all test sites measurements are being made in estuaries where surface waves are fetch-limited and certainly not related to wind speed by the relationship developed by Cox and Munk (1954). $\rho_{as}$ is set to a fixed value of 0.0256 instead of the wind speed formula of Ruddick *et al.* (2006), based on Mobley (1999).





The average and standard deviation of $R_{w1}$, $R_{w2}$ and $R_{w3}$ are calculated.

Finally, an extra white reflectance correction was performed for residual sky glint by subtracting the water reflectance at 1305 nm from all wavelengths. Pure water absorption is very high at this wavelength such that the water reflectance should be zero. Remaining reflectance at this wavelength is assumed to be related to sky glint.

$$R_w(\lambda)new = R_w(\lambda) - R_w 1305$$

The quality control is performed as follows:

- Evaluating the three Ed measurements ($E_{d1}(0+)$, $E_{d2}(0+)$, $E_{d3}(0+)$) for one sampling. If the difference between the Ed measurements is too high (threshold of 5000 was used) the complete sample is removed. A large difference in Ed indicates highly variable light conditions during the measurements.
- The standard deviation of Rw at 750 nm is calculated. When Rw_stdev(750)> 0.01 the measurement is discarded
- After subtracting 1350 nm an inspection is performed of remaining reflection between 1500 and 1700 nm which could be an indication of problems with skylight reflection or skylight which is not completely white. All measurements in this wavelength range with Rw > 0.005 are removed.

The SeaSWIR dataset contains all water reflectance measurements retained after quality control, as well as the downwelling irradiance (average of $E_{d1}(0+)$, $E_{d2}(0+)$, $E_{d3}(0+)$ ), the downwelling sky radiance (average of $L_{sky1}(\lambda)$, $L_{sky2}(\lambda)$, $L_{sky3}(\lambda)$ , $L_{sky4}(\lambda)$ $L_{sky5}(\lambda)$ $L_{sky6}(\lambda)$ $L_{sky7}(\lambda)$ $L_{sky8}(\lambda)$ $L_{sky9}(\lambda)$ ) the total upwelling radiance (average of $L_{sky1}(\lambda)$ $L_{sky2}(\lambda)$ $L_{sky3}(\lambda)$ $L_{sky4}(\lambda)$ $L_{sky5}(\lambda)$ $L_{sky6}(\lambda)$ $L_{sky7}(\lambda)$ $L_{sky8}(\lambda)$ $L_{sky9}(\lambda)$ ) and all standard deviations.

### 4.2 TRIOS water reflectance

The water reflectance is measured with three TriOS-RAMSES hyperspectral spectroradiometers. Two spectroradiometers measure radiance and one measures the downwelling irradiance. The instruments are mounted on a frame, which is projected over the side of the platform using a 2m horizontal tube. Zenith angles of the sea- and sky-viewing radiance sensors are 40°. The azimuth angle of the sensors is adjusted prior to each measurement to obtain relative azimuth angle with respect to sun of 90°, either left or right (depending on which angle has an unobstructed view of the water). Measurements are made for 10 minutes, taking a scan of the three instruments every 10 seconds. The TRIOS sensors measure from 350 to 950 nm, which is a more limited wavelength range compared to the ASD. The sampling interval of the TRIOS is approximately 3.3 nm and the spectral width isabout 10 nm. Position is measured simultaneously by GPS. Data were acquired with the MSDA-XE software and radiometrically calibrated using nominal calibration constants. Calibrated data for $E_d^{0+}$ , $L_{sea}^{0+}$ , and $L_{sky}^{0+}$ are interpolated to 2.5 nm intervals and exported to MATLAB or R for further processing. Data is recalibrated using calibration updates from annual laboratory or factory calibrations. Full details of the data processing, including scan selection and averaging and quality control are described in [Ruddick et al, 2006], including its Web Appendix 1, except that a fixed value of 0.0256 is used for the air-sea interface reflection coefficient in sunny conditions, instead of the wind speed formulation, because of the fetch-limited surface wave field, as explained previously for the ASD data in section 4.1.



### 4.3 Total Suspended Matter concentration

TSM was analysed from water samples collected and filtered in the field. The filters for the TSM analysis (Whatman GF/F filters with a nominal pore size of 0.7 µm) were first prepared in the laboratory before transportation to the test site. The filters were placed in an aluminium plate and ashed for 1 h at 450°C, then they were dried for 12 h at 60°C in an oven. All filters were  - weighted , stored in Petri dishes and transported to the test sites.

At the test sites, water samples were collected in brown bottles just below the water surface. The water (volumes of 3 to 100 ml) was filtered in triplicate through the filters at low vacuum pressure. The exact volume filtered depended on the turbidity of the water as recommended by (Neukermans et al, 2012), which was measured on site as described in section 4.4. Each filter was then rinsed with Milli-Q water (250 ml) andstored in a freezer on site. They were transported in dry ice to the laboratories (LOV and/or VITO). Here they were temporarily stored at -80 °C. Finally, they were dried for 24 h at 60 °C and weighed again in a dry atmosphere.

Systematically the water samples were filtered in triplicates in order to determine the precision of the TSM concentration measurements. After removal of outliers the average of the triplicates was used for further analysis. When TSM was measured by both the VITO and LOV laboratory, also here the average was used.

### 4.4 Turbidity

Turbidity was measured using portable HACH 2100P and 2100QIS turbidimeters as in Nechad et al. (2009). The instrument records turbidity between 0 and 1000 FNU, with a resolution of three significant figures. A 10-ml vial is filled with the surface water sample and illuminated by a light-emitting diode with emission at $860 \pm 60$ nm. The instrument measures turbidity via the ratio of light scattered at an angle of $90° \pm 2.5°$ to forward- transmitted light as compared to the same ratio for a standard suspension of Formazin. This optical measurement technique of turbidity is in accordance with ISO 7027 (1999), and determines turbidity in Formazin Nephelometric Unit (FNU). Turbidity was recorded in triplicates that were averaged. Turbidities of the STABLCAL Stabilized Formazin Turbidity 10 or 20, 100 and 800 FNU standards and that of pure water were recorded after each sampling campaign to check the instrument stability.

### 5. Results and discussion

### 5.1 ASD and TRIOS reflectance

The SeaSWIR reflectance dataset is generally characterized by low reflectance values at short wavelengths (< 450 nm), peak reflectance values between 600 and 720 nm and significant contributions in the NIR and SWIR-I parts of the spectrum – see Figure 5. However significant variations exist among the different test sites. To further explain these differences, the water reflectance spectra for all data campaigns are displayed in Figure 5 from 350 to 1300nm. Beyond 1300 nm the pure water absorption is extremely high (>140 m$^{-1}$) and there is no reflectance emerging from the water. In Figure 5, 5 characteristic spectral regions are defined between 350 and 1300 nm: A) between 350 and 600 nm, B) between 600 and 720 nm, C) in the NIR between 720 and 1000 nm, and finally regions corresponding to D) SWIR-I (1000 - 1200 nm) and E) SWIR-II.




The spectra for the Scheldt (data campaigns Scheldt2010 and Scheldt2012, Figure 5a) and 5b) have overall high reflectance values. In region A the spectra have a similar slope related to light absorption of CDOM and non-algal particles. In region B, maximum reflectance values are found up to 0.1 and the spectra show clear changes in shape and magnitude. The influence of light absorption by Chlorophyll-a, for example, is clearly noticeable around 665 nm in the Scheldt2012 dataset.  In region

C, there is more variability in the magnitude of the spectra more related to the concentration of particles while spectral variability is determined by pure water absorption [Ruddick et al, 2006]. In SWIR-I a significant signal can be observed with maxima around 0.02 (at 1071 nm). In SWIR-II the maximum reflectance is 0.001 at 1268 nm. The shape of the water reflectance spectrum in the SWIR is also clearly influenced by pure water absorption [Knaeps et al, 2012].

The spectra for La Plata are shown in Figure 5e. All the spectra have a very similar shape and have only small differences in magnitude. Maximum reflectance values are found around 700 nm with values up to 0.1. There is also a local minimum around 665 nm probably related to chlorophyll-a absorption. There is a very weak signal in SWIR-I (maximum 0.007 at 1071 nm), while no significant signal in SWIR-II.

The Gironde spectra (data campaigns Gironde2012 and Gironde2013, Figures 5c and 5d) have extremely high values, up to 0.2 around 700 nm. Also here, a saturation effect is observed at the shorter wavelengths in region A, followed by a spectral region with high reflectance (B) with maximum values at 500 and 700 nm.  In region (C) very large variability in reflectance values are observed. A reflectance signal up to 0.07 is observed in SWIR-I and a small signal is observed in SWIR-II (highest reflectance signal is 0.0028 at 1276 nm). For the MODIS and VIIRS bands at  1240 nm, a maximum reflectance of

0.0026 is measured. Two measurements have $R_w(1240)$ above 0.002 and 6 measurements have $R_w(1268)$ above 0.002.

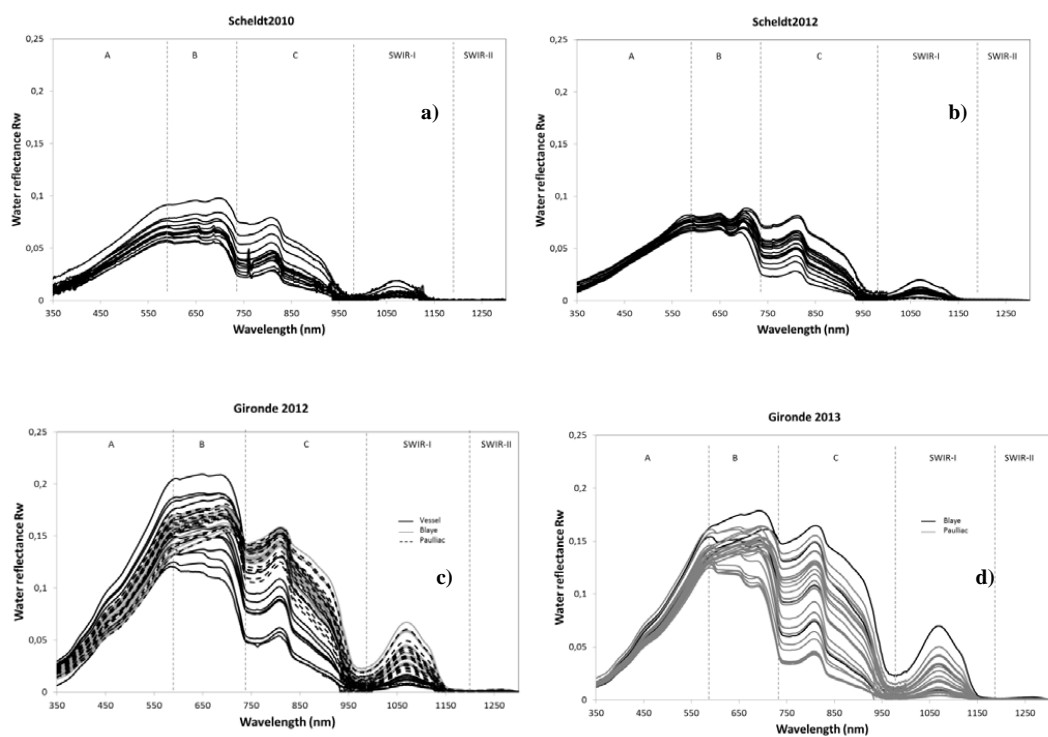



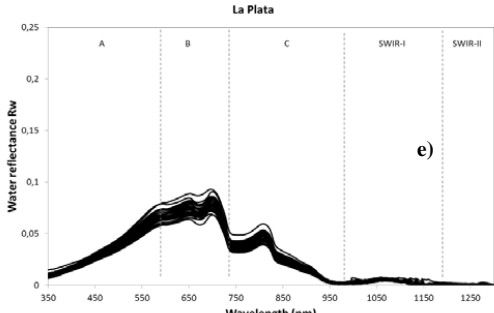

**Figure 5: ASD water reflectance for all data campaigns**

To compare the reflectance spectra from the three estuarine sites, all reflectance spectra with a TSM concentration around

100 mg L$^{-1}$ for all 3 sites are plotted in Figure 6 (exact concentrations: 96 and 105 mgL$^{-1}$ for the Gironde; 91, 93, 96 and 110 mgL$^{-1}$ for La Plata; 102, 107 and 110 mgL$^{-1}$ for the Scheldt river). The mean and standard deviation (stdev) from all spectra are calculated at 6 different wavelengths (510, 650, 865, 1020, 1071 and 1240 nm) and are presented in Table 3. From the figure it can be observed that the reflectance spectra in A and B spectral regions are very different (stdev/mean = 0.298 and 0.275, respectively), mainly due to the high reflectance values observed for the Gironde. Further, in region B the reflectance

spectra for the Scheldt are flatter than the reflectance spectra for Río de la Plata, which gradually increase to a maximum reflectance around 695 nm. All spectra exhibit a similar shape above 700 nm clearly related to the pure water absorption. For the same TSM concentration the reflectance spectra are very similar at 865 and 1071 nm (stdev/mean = 0.077 and 0.13, respectively). Finally, a clear reflectance signal is observed for all sites in SWIR-I, but no significant signal for SWIR-II can be detected. The reflectance increase observed around 1275 nm for one spectra from Río de la Plata is treated as a

measurement error.

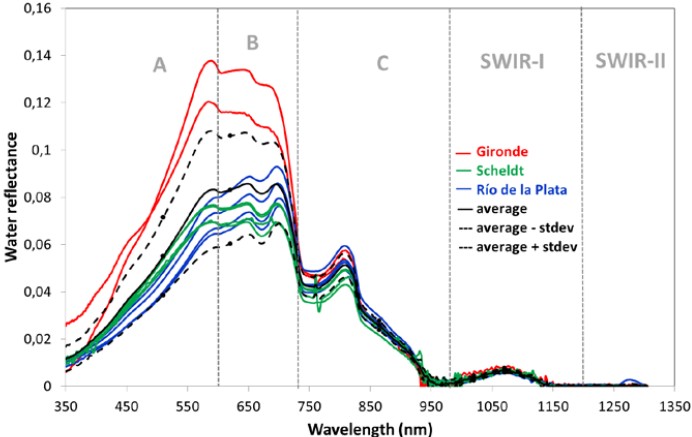

**Figure 6: Water reflectance spectra with a TSM concentration of ~100 mg L$^{-1}$ for the Scheldt (n=4), Río de la Plata (n=4) and Gironde estuaries (n=2). The average of all spectra is shown as a black solid line. The average minus and plus the standard**

**deviation is shown in dashed black line.**

**Table 3: Statistics for the ASD reflectance for the reflectance spectra shown in Figure 6 with a TSM concentration of ~100 mg L.**



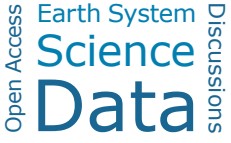

| Wavelength (nm) | Spectralregion | Mean | Stdev | Stdev/Mean |
|---|---|---|---|---|
| **510** | A | 0.0550 | 0.0164 | 0.2983 |
| **620** | B | 0.0831 | 0.0229 | 0.2754 |
| **865** | C | 0.0257 | 0.0020 | 0.0773 |
| **1020** | SWIR-I | 0.0035 | 0.0011 | 0.3075 |
| **1071** | SWIR-I | 0.0067 | 0.0009 | 0.1319 |
| **1240** | SWIR-II | 0.0001 | 0.0005 | 3.1443 |

## 5.2 Intercomparison ASD-TRIOS

For all stations where both TRIOS and ASD reflectance is available, the correlation is shown in Figure 7 for 6 spectral bands. The best correlation is obtained for the shortest wavelengths (412, 490 and 555 nm) with a correlation coefficient of 0.98. The correlation coefficients for the longer wavelengths (683, 780 and 865 nm) is slightly lower (0.93) and more scatter is observed. Again, we can observe slightly higher water reflectance values for the TRIOS. Overall these results are very satisfying and give confidence in the ASD measurement methodology and processing.

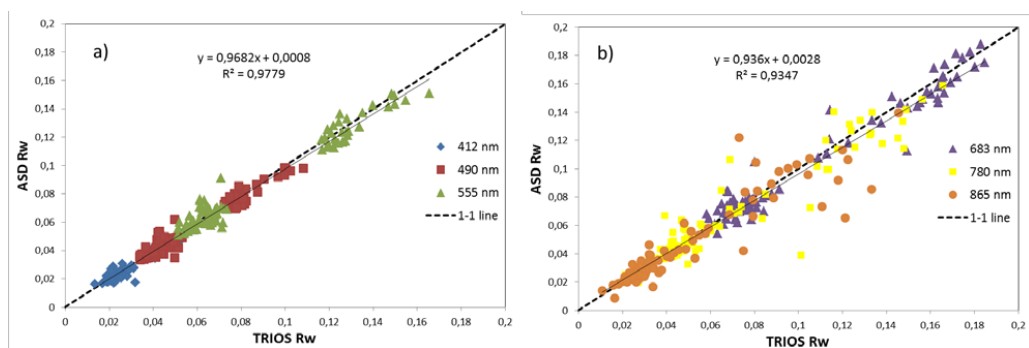

**Figure 7: ASD water reflectance versus TRIOS water reflectance**

## 5.3 TSM and Turbidity results

The mean, minimum, maximum and standard deviation of the TSM concentrations and turbidity measurements are shown in
Table 4 and Figure 8. Based on the triplicates, the accuracy was excellent (<5% on average) for the Scheldt and the Gironde and ~10% for La Plata. When interpreting these results, it should be noted that for the Gironde 2012 campaign, only limited turbidity measurements were available (only for last measurement day in Pauillac).

TSM concentrations measured over the three sites range from 48 mg L-1 up to extreme values of 1400 mg L-1. Turbidity ranged from 48 to 1422 FNU (calculated with dilutions). At the pontoon of Pauillac, the highest TSM concentrations were
recorded and the lowest were recorded at La Plata. The TSM concentration in the Gironde follows the tidal cycle and can vary one order of magnitude in one day (standard deviation of 294 and 269 mg L-1 for Gironde 2012 and Gironde 2013, respectively). Little variation was observed in La Plata with a TSM standard deviation of 15 mg L-1.

**Table 4: TSM and turbidity**

| | TSM min (mg L-1) | TSM max (mg L-1) | TSM mean (mg L-1) | TSM stdev (mg L-1) | T min (FNU) | T max (FNU) | T mean (FNU) | T stdev (FNU) |
|---|---|---|---|---|---|---|---|---|
| Scheldt2010 | 55,0 | 402,0 | 146,3 | 106,2 | 52,3 | 282,0 | 115,6 | 73,7 |
| Scheldt2012 | 49,6 | 300,7 | 166,2 | 70,2 | 48,1 | 331,0 | 166,8 | 74,3 |





| Gironde2012 | 96,0 | 1400,5 | 452,9 | 294,4 | 371,5 | 807,3 | 540,2 | 175,2 |
| La Plata 2012 | 48,3 | 110,0 | 70,7 | 15,0 | 65,2 | 127,0 | 90,0 | 11,9 |
| Gironde2013 | 86,3 | 1190,0 | 368,4 | 269,1 | 66,4 | 1422,5 | 424,4 | 330,2 |

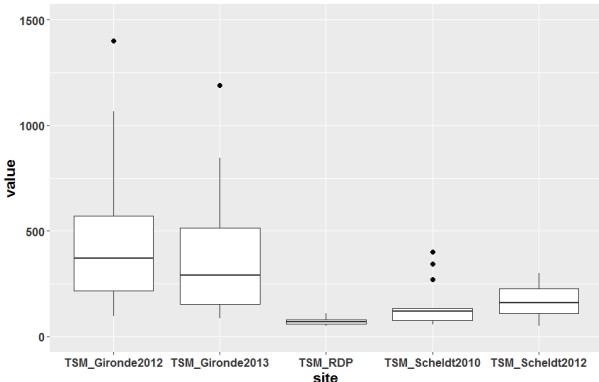

**Figure 8: Boxplot for TSM for all sites**

A few water samples from the Gironde and Scheldt campaigns were analyzed for TSM at both the VITO and LOV laboratories. Figure 9 show the relationship between both. The correlation is high ($R^2$ is 0.96) and there is only a very small offset of 0.8 mg L-1.

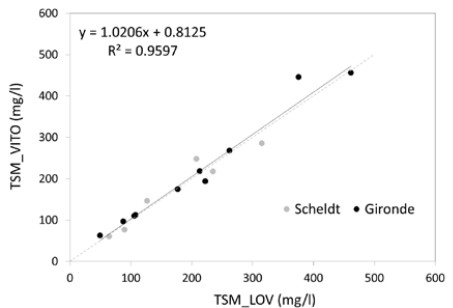

**Figure 9: Intercomparison of TSM measured by the different laboratories.**

**5.4 Intercomparison TSM-turbidity**

The TSM-turbidity relationship for all sites is linear with a correlation coefficient of 0.96 (Figure 10a). The outlier observed at 456 FNU and 176 mg L-1 TSM is station gir61 from the Gironde 2012 campaign (second measurement day at Pauillac). The trendline is slightly steeper than the 1-1 line and there is a negative offset of -9.7 mg L-1. Figure 10b shows the TSM-
turbidity relationship with the different sites in different colours and Figure 11c zooms in on the data with TSM concentration below 300 mg L-1. Slightly different relationships can be observed for the different sites with the Gironde trendline matching more closely the 1-1 line and the Scheldt and la Plata trendlines more deviating. This can be related to the different particle size and composition at the sites and suggests that different relationships are needed when converting T to TSM.



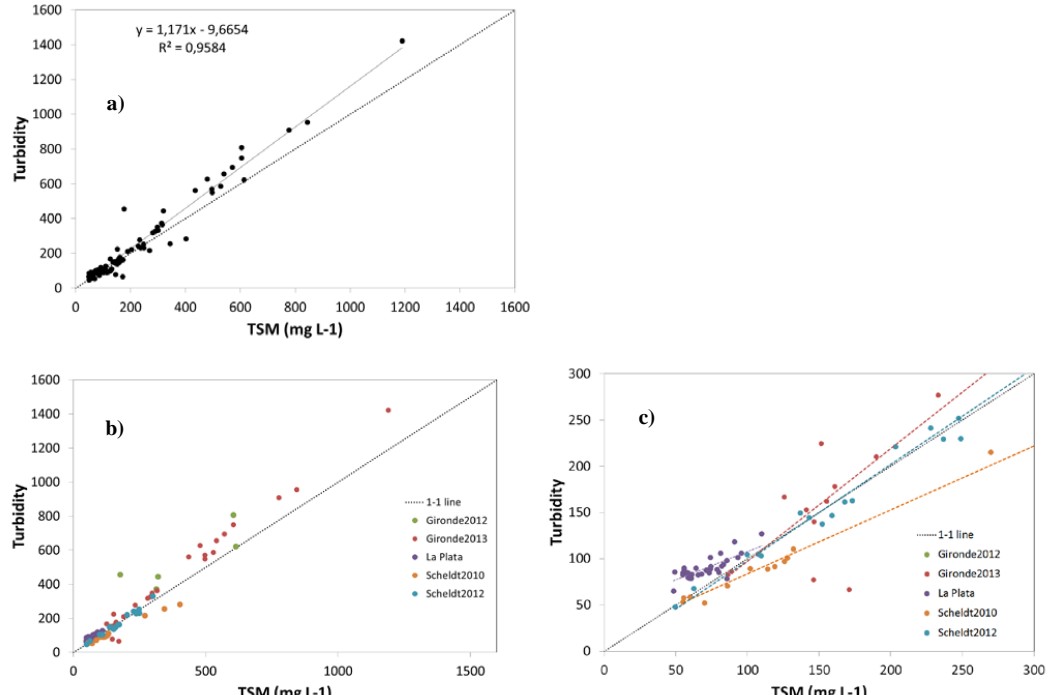

**Figure 10: Intercomparison TSM and turbidity**

### 6. Conclusions

The SeaSWIR dataset presents the first hyperspectral measurements of the water reflectance in the SWIR. Measuring water reflectance in the SWIR was made possible by using an ASD spectrometer with one radiometer and a measurement protocol where the instrument is turned facing the sky, the water and the spectralon panel. Using this measurement protocol and consecutive post-processing it was shown, by intercomparison with TRIOS measurements, that high quality water reflectance measurements can be made with the ASD. It is however suggested to strictly follow the same procedure as

outlined in this paper and apply a strict quality control. Measurements which passed quality control were all performed under stable atmospheric conditions, being blue sky or completely overcast. Variable light conditions prohibit a correct retrieval of the water reflectance as lighting conditions change while turning the instrument and making the three individual measurements of downwelling irradiance, the total upwelling radiance and downwelling sky radiance. It should also be noted that most measurements were made from fixed pontoons, which simplifies the measurement significantly and

improves the quality of the data. Still, even when measuring on a pontoon, the location on the pontoon should be chosen wisely and should not be close to any large obstacles which could alter the measurements of the downwelling irradiance and downwelling sky radiance.

The quality of this dataset opens up the possibility of including the ASD in future measurement campaigns on water, for the development of algorithms as well for calibration and validation of satellite and airborne observations. This is important for

current (e.g. Sentinel-3 OLCI) and future satellite missions (e.g. Environmental Mapping and Analysis Program - EnMAP, Hyperspectral Precursor and Application Mission – PRISMA and Sentinel-10) with more spectral bands in the SWIR-I and SWIR-II.

Next to the water reflectance measurements, the SeaSWIR dataset also contains simultaneous TSM and turbidity measurements (using a HACH portable turbidity meter). The high quality of these measurements was confirmed by the



standard deviation of the triplicates and the duplicate measurements performed in the LOV and VITO laboratories. As all measurements were made in estuaries during the tidal cycle, the TSM range of the SeaSWIR dataset is very high ranging from 48 mg L-1 up to extreme values of 1400 mg L-1. Hence, the SeaSWIR dataset is not only unique in terms of its spectral content but also in terms of its concentration range.

**Acknowledgement**

The research leading to these results has received funding from the Belgian Science Policy Office through the STEREO program (SeaSWIR project)

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
