# Peer review of "The SeaSWIR dataset"

_Earth System Science Data, 2018_

## Referee Comment (RC1) · K. Voss (Referee) · 23 May 2018

General comments:

This paper describes and presents a unique data set which combines turbidity, total Suspended matter (TSM), and radiometry (reflectance) extending from the visible to the short wave infrared wavelenths up to 1300 nm. The measurements are from highly turbid environments, with a range of TSM from 48 to 1400 mg/L, and in a few different kinds of environments, to vary the other water properties. It is a unique data set, because of the radiometric measurements in the infrared, and should be very useful to those working on both TSM algorithms, and atmospheric correction in turbid environments. The paper is well written and generally clear.

Specific comments:

[Figure]

I didn't have any specific comments on the overall scientific questions, as always it would be interesting to have more parameters with the data set (some sort of CDOM parameter, scattering and absorption at visible wavelengths). But the data set as is, will be very useful for those working in these types of environments.

Technical comments:

Page 1, line 27: I think it should be ". ...because specialized optical. ...

Page 2, line 13: I think it should be analyzes, rather than analyses...

Page 3, line 15: suggest "allow measurement of a large variation. ..."

Page 3, line 16 and 17: rather than "verticality" and "horizontally", how about vertical orientation and horizontal orientation. ...

Page 5: line 10 and 11: Figure 2 doesn't show an ASD in black pointing at the spec-tralon plaque.

Page 7, line 7: 5000 what? While the units maybe arbitrary, can you give the threshold in terms of some percentage or in some way inform the reader as to how big a variation 5000 is in terms of Ed?

Page 7, line 12: What do you mean by "skylight is not completely white"? Why would you expect it to be?

Page 7, line 28: space between "is" and "about"

Page 9, line 13: replace "while" with "and"

Page 12, line 14 and vicinity: TSM and T are in different units (mg/L and FNU). Why would you expect a 1 to 1 line? The negative offset might be interesting, but I wouldn't expect a 1:1 line necessarily between two things with different units...

Page 13: figure 10, should have units for Turbidity (FNU).

Throughout, the figure captions could use some work to more fully explain the figures.

[Figure]

At least the what the different panels are showing should be described in the caption, to make them clearer without having to scramble through the paper. . .in which the order is not necessarily always from a-e or whatever. . ..

Also, in the reflectance data, it would be helpful to have a separate column for wavelength to make it easier to plot quickly. Probably picky, but easier than having to parse out the wavelength from the column description. . ..

---

## Referee Comment (RC2) · Interactive comment on "The SeaSWIR dataset" by Els Knaeps et al. · 28 May 2018

Els et al presented an important dataset for the development, validation, and evaluation of algorithms for suspended particulate matter (SPM). In particular, this dataset covers >2 orders of SPM magnitude, along with the state-of-the-art measurement of water's reflectance. I strongly support its publication, sharing, with the broad community, after a few minor polishing of the manuscript. Specifically, 1. Page 1, Line 16, change "L-1" to "L-1" with "-1" in the superscript. It happens to many places. 2. P1, Line 24, insert space between "1." and "Introduction". 3. P1, Lines 29-32 "More information . . ." is a long sentence, please check grammar to make this sentence clear. 4. P2, Lines 1-4, it appears this sentence is out of sync of the previous sentences. 5. P2, line 17, insert space between "2." and "Sites". 6. Remove the extra punctuations in lines 19-21. 7. P5, line 15, downwelling radiance → downwelling irradiance. 8. P7, line 1, "Rw2 and Rw3", the "and" should not be subscript. Please add a sentence to clarify Rw2 and Rw3. 9. P7, line 10, "threshold of 5000". What is this 5000? 10. P8, line 35, "5

characteristic ..” → “five characteristic ..”. 11. P11, Fig. 7. Please use two significant digits for the R2 values. Same for Fig. 9, Fig. 10. 12. P12, Fig. 8. Please use more appropriate title for the Y axis. 13. P12, Fig. 9, units, mg/l → mg/L.

---

## Author Comment (AC1) · 26 Jun 2018

We would like to thank the referees for their comments and suggestions.

We have taken into account the grammatical suggestions and also changed the figures according to the comments.

We did a small re-organization of the introduction text as requested by the referees.

We added an explanation on the white correction. This white correction was applied because several measurements were performed under homogeneous cloudy conditions. It was observed that the skyglint is relatively white under these circumstances and a additional white correction improves the results.